# Investigating the Prevalence of Paratuberculosis in Hungarian Large-Scale Dairy Herds and the Success of Control Measures over Four Years

**DOI:** 10.3390/ani14010151

**Published:** 2024-01-02

**Authors:** Barbara Vass-Bognár, Johannes Lorenz Khol, Walter Baumgartner, Kinga Fornyos, Melitta Papp, Zsolt Abonyi-Tóth, Mikolt Bakony, Viktor Jurkovich

**Affiliations:** 1Department of Animal Hygiene, Herd Health and Mobile Clinic, University of Veterinary Medicine, H-1078 Budapest, Hungary; bognar.barbara@univet.hu; 2University Clinic for Ruminants, Department for Farm Animals and Veterinary Public Health, University of Veterinary Medicine, 1210 Vienna, Austria; 3Eurofins Vetcontrol Ltd., H-1211 Budapest, Hungary; 4Department of Biostatistics, University of Veterinary Medicine, H-1078 Budapest, Hungary; 5Centre for Translational Medicine, Semmelweis University, H-1085 Budapest, Hungary; bakony.mikolt@semmelweis.hu; 6Centre for Animal Welfare, University of Veterinary Medicine, H-1078 Budapest, Hungary

**Keywords:** paratuberculosis, seroprevalence, dairy cattle, test-and-cull, biosecurity

## Abstract

**Simple Summary:**

Paratuberculosis is an incurable disease of cattle that causes serious economic damage. The only protection method is to prevent the spread of infection within the farm by culling the infected animals. We examined various control measures regularly carried out on 42 farms over four consecutive years (2018–2021), and how effective they were in reducing the occurrence of paratuberculosis. We found that farms that base their strategy only on blood testing and culling infected animals cannot sufficiently reduce the incidence of the disease. It is necessary to stop the spread of the infection by preventing the calves from coming into contact with infectious material.

**Abstract:**

Paratuberculosis (PTB) is a severe, slow-developing, untreatable disease of ruminants. Worldwide, the disease affects more than 50% of herds in the dairy industry, and causes substantial economic losses for dairy producers. Diagnostic tests show limited sensitivity, especially in the early stages of the disease. Our study aimed to investigate the seroprevalence of *Mycobacterium avium* ssp. *paratuberculosis* (MAP) in large-scale dairy herds in Hungary, in association with the self-reported presence or absence of screening and intervention measures against MAP transmission. We processed data from 42 large-scale Holstein Friesian farms in Hungary between 1 January 2018 and 31 December 2021. An average of 32,009 (min.: 31,702; max.: 32,207) animals were blood sampled yearly (127,372 in total during the four years), corresponding to 15% of the Hungarian dairy cattle population. All female cattle older than 2 years were blood sampled on the farms enroled in the study. The samples were tested using a commercial ELISA (IDEXX paratuberculosis screening Ab test). Farm managers were interviewed about their on-farm diagnostic and intervention approaches using a uniform questionnaire, including questions on the level of awareness, frequency of ELISA and PCR testing, and their strategies for culling adult animals and reducing transmission to newborn calves. By comparing the annual rate of change in seroprevalence and the amount of change observed during the four-year period, we concluded that test-and-cull strategies implemented in parallel with newborn calf management that aimed at preventing MAP transmission were superior to test-and-cull strategies alone; moreover, fortifying culling decision making via additional ELISA and PCR tests is superior to using a single ELISA result. For farms that carried out a complex program with both “test-and-cull” and proper newborn calf management, there was a proportional reduction in apparent seroprevalence at an average of 22.8% per year. Fifteen of the sampled farms had no measures in place to control paratuberculosis. On these farms, the seroprevalence increased by 12.1% per year on average.

## 1. Introduction

Paratuberculosis (PTB) is a severe, slow-developing, untreatable granulomatous enteritis caused by the *Mycobacterium avium* subspecies *paratuberculosis* (MAP) [1], which occurs in domestic and wild ruminants [2]. Paratuberculosis has a long incubation period, after which the clinical effects worsen in severity, from diarrhoea and reduced milk production to lethargy, hypoproteinaemia and severe emaciation [3]. Worldwide, the disease affects more than 50% of herds in the dairy industry [4], and causes substantial economic losses for dairy producers [5]. The primary sources of losses are decreased milk production, decreased slaughter value and premature culling [6,7,8,9]. MAP may also have a role in human Crohn’s disease. Crohn’s disease is thought to result from interactions between environmental and genetic factors and persisting antigens [10]. Due to its direct effect on animal health, potential zoonotic risk and economic losses, the disease has been listed by the World Organization for Animal Health (WOAH [11]). According to European regulations, PTB is a “category E disease”, meaning that there is a need for surveillance within the Union [12].

In most cattle populations, the prevalence of PTB is significantly higher in large-scale herds compared to small family farms [13]. It is difficult to estimate the true prevalence of PTB worldwide, as different diagnostic methods are used and the population sizes studied differ greatly in each country [5]. Additionally, diagnostic tests show a limited sensitivity, especially in the early stages of the disease [14,15]. A survey of European data between 2010 and 2017 showed that in countries where paratuberculosis cases were regularly reported to the WOAH, seroprevalence increased significantly over the 8-year surveillance period by an average of 0.6% per year [16]. Based on testing MAP antibodies from milk at the time of milk recordings, the latest survey in Hungary in 2020 shows that PTB is an increasing problem in large-scale dairy herds. In 2020, the true prevalence at the herd level was 89.1%, and at the animal level it was 4.4% for primiparous and 10.3% for multiparous cows [9].

Whittington et al. [17] conducted a narrative review of 48 countries (2012–2018) on PTB control programs. The authors reported that some countries have obligatory PTB eradication programs, such as Sweden and Austria, where clinical paratuberculosis is a notifiable disease [18]. In contrast, others (e.g., Spain) have voluntary programs on a region-by-region basis, and only 46% (22/48) of countries had an established control program for the disease. In Hungary, PTB ELISA tests were state-supported until 2022.

The main difficulties in control programs for paratuberculosis eradication are the long incubation period of the disease; the different and impaired sensitivities of commercially available diagnostic tests; and the long survival of MAP in the environment [19]. A key element of control programs to reduce infection pressure is the culling of infectious animals from the herd [20]. Another key aspect is hygiene and management control. Preventing calves from being exposed to the faeces of adult animals and giving colostrum and milk to calves only from MAP-negative animals also play very important roles in interrupting the infectious chain. PTB is one of many infectious diseases that spreads widely due to poor hygiene [21]. Two main groups of control strategies can be differentiated: programs based on testing and culling of positive animals (“test-and-cull” [22]) and programs involving hygiene and management actions, or a combination of these strategies [23]. For herd-level diagnostics, serum and milk ELISA samples provide the most rapid results [24]. Still, in recent years, faecal qPCR tests have seen widespread use, as they allow the detection of bacterial shedding before the onset of the humoral immune response [25]. In many cases, control programs are hampered by the cost, energy and time involved for veterinarians and farmers [26,27]. Although paratuberculosis eradication programs have been set up for almost a century, the dairy sector has not yet developed a uniform protocol that can be applied consistently with a high success rate [21]. 

Our study aimed to investigate the prevalence of paratuberculosis in Hungarian dairy herds. The objective was to monitor the progress of control programs in dairy herds, and to observe herds where no measures to control paratuberculosis are in place. Our study also aimed to present trends in Hungary and compare the success of control programs on Hungarian farms, both with each other and using data from international literature. The overall aim was to develop recommendations that can be generally applied to supporting farms to start and implement control programs for paratuberculosis in dairy cattle.

## 2. Materials and Methods

### 2.1. Farms and Animals

We processed data from 42 large-scale (with a population of at least 200 dairy cows) Holstein Friesian farms in Hungary between 1 January 2018 and 31 December 2021. The average number of dairy cows per farm was 739 (min.: 208; max.: 2064; standard deviation: 459). In Hungary, the cost of paratuberculosis antibody testing was subsidised by the state until August 2022, but at the same time, the testing was voluntary. Test results could serve as a basis for PTB control strategies. The tested farms in our study voluntarily participated in the subsidised tests and agreed to complete a questionnaire on PTB control strategies. The farms were not selected randomly from all Hungarian farms; the basis of selection was access to the results of herd-level ELISA, having been performed by the laboratory to which the first author has work connections. However, the farms included show great variability in geographic location and cow population, which makes our sample representative of local conditions.

### 2.2. Antibody Testing

Blood samples were taken by farm veterinarians from the cows’ coccygeal vessels into native sampling syringes (Monovette; Sarstedt AG & Co., Nümbrecht, Germany). The syringes were sent to the laboratory after sampling. An average of 32,009 (min.: 31,702; max.: 32,207) animals were sampled yearly, corresponding to 15% of the Hungarian dairy cattle population. All female cattle older than 2 years were tested on the farms enroled in the study. Serological testing was performed on the animals’ serum samples using the IDEXX paratuberculosis screening Ab test (IDEXX Laboratories, Inc., Westbrook, ME, USA) ELISA kit. With this kit, optical density (OD) values were transformed to S/P ratios based on the OD for the serum sample, together with those for the negative and positive controls provided with the kit, using the following equation: S/P ratio = (OD of sample − OD of negative control)/(OD of positive control − OD of negative control) [24]. All of the assays were run in duplicate. According to the manufacturer’s instructions, the cut-off values were as follows: negative S/P < 45%; positive: S/P > 55%. Considering the specificity and sensitivity characteristics of the ELISA, we used the term apparent prevalence, defined as the portion of tested animals with a positive test (T+). Therefore, pr = P (T+) = TPR + FPR, where TPR and FPR are the true-positive and false-positive rates, respectively [28]. Hereafter, where prevalence is concerned, we mean apparent prevalence.

Only laboratory results from Eurofins Vetcontrol Ltd. (Budapest, Hungary) were used to ensure data comparability. To calculate the annual seroprevalence, the data of the MAP test performed in parallel with the compulsory annual brucellosis herd blood sampling were used. Results from other additional paratuberculosis tests performed during the year were not included in the seroprevalence calculation to avoid biasing by non-equal or possibly repeated sampling carried out on the farm. 

Faecal PCR examinations were implemented in some farms to support their culling decision. In these cases, the Adiavet ParaTB real-time kit (Bio-X Diagnostics S.A., Rochefort, Belgium) was used by the lab for the qPCR run. The culling decision was made based on the PCR Ct values; however, there was no exact threshold for “high shedder” animals. The MAP shedding rate was always evaluated by the farm vet, and it was compared to the previous herd averages.

### 2.3. Measuring the Paratuberculosis Control

The tested farms were contacted every year by telephone or in person, and they were asked about their awareness of PTB and what screening and intervention measures were implemented on the farm, if any. We used a uniform questionnaire on all farms containing the same yes/no and open-ended questions. The questions referred to awareness, in general and more specifically, to diagnostic and intervention measures as follows: (1) *PTB awareness*: (a) no targeted control is implemented or (b) at least some form control measure is in action; (2) *screening*: (a) only the annual subsidized herd-level serology is performed, (b) the subsidized test is complemented with individual ELISA testing or (c) beyond the subsidized and individual ELISA, individual PCR tests are also occasionally performed; (3) *intervention*: (a) no intervention, (b) planned culling based on test positivity and (c) planned culling based on test positivity and perinatal preventive measures. In the open-ended questions, we asked about the culling strategy in more detail, namely the basis of the decision and the timing. Also, we inquired about what specific perinatal preventive measures were implemented, e.g., the separate calving of PTB+ cows, the immediate removal of calves from their mothers in the case of PTB-positive animals and feeding colostrum and raw milk originating from only MAP-negative animals [29,30]. 

Farms, where the answers were inconsistent across the four years, were excluded from the analysis. 

### 2.4. Statistical Analysis

Mixed Poisson regression models were used to examine the effects of different control methods over time on seroprevalence. Since individual effects of farms could not be eliminated, a mixed model was used. In this model, the dependent variables were the number of seropositive cows, and the offset was the logarithmic sample size. Univariate analyses were performed, in which the fixed factors were time in years, the yes/no answers about PTB awareness, screening and intervention methods, as well as their interactions with time. Due to the high variation in culling strategies used, the answers to the open-ended questions could not serve as a basis for statistical exploration. Farm effects were considered with individual slopes and intercepts. The annual change in seroprevalence was calculated using contrasts. 

In a second analysis, six levels of PTB control were defined by combining the levels of screening and intervention. The percentage point changes in the seroprevalence over the four-year timespan were calculated as a summary measure of the efficiency of control interventions. Then, these percentage point changes were compared between the different levels of PTB control using the Welch F-test (ANOVA without the homoscedasticity assumption). Post hoc comparisons were made with the Dunnett test, using “herd-level ELISA and no intervention” as the reference category.

Baseline comparability was assessed in both statistical analyses by comparing initial seroprevalence across levels of the studied variable. The level of statistical significance was *p* < 0.05. All statistical calculations were performed using the statistical software R 3.2.3 [31].

When reporting descriptive statistics, percentages (%) refer to seroprevalence (number of test-positive cows/total number of cows) and percentage points (pp) refer to the *amount* of change in seroprevalence (change calculated by subtraction). When reporting estimated effect sizes, percentages (%) refer to the annual *rate* of change in seroprevalence (change calculated by division).

## 3. Results

The numbers of included farms according to the different aspects of the questionnaire are displayed in Table 1. 

Not taking the differences in the attitude towards PTC control into account, the overall average of apparent PTB seroprevalence across all farms studied was 5.1% in 2018, and 5.6% in 2021. Baseline comparisons indicated no significant difference in initial seroprevalence between farms according to neither PTB awareness, screening or intervention levels (*p* = 0.6587, *p* = 0.5351 and *p* = 0.766, respectively). This way, the baseline seroprevalence did not bias our results.

The interaction between time and each studied variable was significant (*p* < 0.0001 in all cases). This indicates that the slopes of the lines describe the changes in seroprevalence over time; that is, the annual rate of change in seroprevalence differs between farms that answered yes from those that answered no to the given question. The effect sizes estimated by the mixed Poisson regression models are displayed in Table 2. The reported *p*-values refer to the general linear hypothesis tests on the effect sizes differing from null.

The absence of any intervention measures (PTB awareness: no) was significantly associated with an average 12% annual increase in seropositivity (for illustration, over 6 years, it means that seroprevalence doubles). Also, in the more specific aspects, an answer of no was significantly associated with a worsening of the PTB situation. Performing complementary diagnostic tests was associated with a decrease, while the absence of such was associated with an increase in seroprevalence. Effect sizes associated with intervention measures suggest maintenance or decrease in herd-level seropositivity.

In the case of the farms where only a planned culling strategy was carried out, there was no significant difference in the year-to-year change in seroprevalence (−4.6%; 95% CI: −10%; 1%; *p* = 0.150). For the farms that carried out a complex program with planned culling and perinatal prevention strategies, there was a proportional reduction in apparent seroprevalence of 23% per year (95% CI: −28%; −12.0%; *p* < 0.001).

Table 3 displays answers to the open-ended questions regarding the basis and timing of the culling decision, and the corresponding achieved percentage point changes in seroprevalence. The small sample sizes did not allow for statistical comparisons.

Our results show that following only the “test-and-cull” concept—in any of its various forms—does not necessarily lead to a reduction in seroprevalence, as shown by the 0.2 pp change in seroprevalence (min.: −5.1 pp; max.: +3.5 pp; standard deviation 1.8 pp). One farm among these farms was an exception, achieving a 5.1 pp reduction in seroprevalence compared to its initial values. This farm used additional serology, sampling twice yearly (around 10–14 days after calving and drying off) in addition to the annual testing. Animals with high S/P were immediately culled if not pregnant, and the remaining were not inseminated. This constitutes more frequent testing with a more stringent culling protocol than the traditional “test-and-cull” method. To facilitate the culling decision, this farm often performed additional diagnostic tests as part of its protocol, such as additional ELISA or faecal PCR tests.

The estimated effect sizes served as a basis for creating ordered categories that define the level of awareness. In terms of screening, performing complementary ELISA was shown to be superior to only herd-level ELISA, while additional PCR was shown to be superior to complementary ELISA. In terms of intervention, planned culling was shown to be superior to no intervention, while additional preventive calving measures were shown to be superior to planned culling. Also, added efforts in intervention were shown to be superior to added efforts in screening. Based on such rankings, the following order of awareness categories were defined:Herd-level ELISA and no intervention;Herd-level ELISA and planned culling;Herd-level and individual ELISA and planned culling;Herd-level and individual ELISA + individual PCR and planned culling;Herd-level and individual ELISA and planned culling + perinatal prevention;Herd-level and individual ELISA + individual PCR and planned culling + perinatal prevention.

Using these categories in the second analysis, we compared the amount of change in seroprevalence observed from the start to the end of the four-year study period. The initial and closing seroprevalences according to the awareness category are shown in Figure 1.

While initial seroprevalences were similar across categories, closing seroprevalences showed greater variation and an apparent decrease, with an increasing level of awareness. 

Table 4 shows mean (±SD) initial and closing seroprevalences and the amounts of change during the four-year timespan across awareness categories.

The mean baseline (2018) seroprevalence showed no significant difference across categories (*p* = 0.7723). However, the mean closing (2021) seroprevalence was significantly associated with the category (*p* < 0.001). The post hoc Dunnett’s test indicated that the mean prevalences in categories 5 and 6 significantly differed from reference category 1 (*p* = 0.0270 and 0.0106, respectively.). The mean percentage point difference was also significantly associated with category (*p* = 0.0061). The post hoc Dunnett tests indicated a significant difference between categories 4, 5 and 6 with respect to reference category 1 (*p* = 0.0065, *p* < 0.001 and *p* < 0.001, respectively). The mean percentage point differences were also compared between the categories, excluding category 1 (herd-level ELISA + no intervention level). That is, only the farms implementing at least some control measures were compared. No significant associations were found between category and mean change in prevalence (*p* = 0.1246). 

## 4. Discussion

The average annual increase in the overall paratuberculosis seroprevalence over the four years in the surveyed farms was just above 0.5 pp. This result is in line with the study by Fanelli et al. [16], who reported an annual increase in the overall seroprevalence of 0.6% over eight years in European countries where paratuberculosis was regularly reported. We could not compare the four-year data to the Hungarian average, as this is the first longer-term study about the change in PTB seroprevalence. 

Of the surveyed farms, 29 had some control program for PTB in place. However, in most farms, this was only sufficient to keep the disease prevalence from increasing rather than substantially reducing it. According to the literature, it is advised to judge the success of most PTB control programs after five years [32]. Unfortunately, it was not possible in the present study to analyze 2022 data with the end of the state subsidy in 2022; most farms did not continue to perform serological tests on a herd level; this way, the level of screening that served as a basis for culling decisions changed considerably. However, there are recent studies [33] that evaluated the success of the programs after four years.

As the present study is an observational one, with limited possibilities of matching, the observed associations have a low level of evidence in terms of causality. However, observational studies provide the opportunity to explore relationships between parameters, and provide grounds to generate hypotheses to be tested in future experiments. 

It can be assumed from our study that the success of PTB mitigation depends largely on the success of preventing infection in calves, suggesting that a “test-and-cull” strategy alone will generally not achieve significant results in the long term. This finding aligns with several publications that report that methods based on testing and culling alone are not necessarily effective in controlling PTB in the long term [34,35]. In our study, the culling of infected animals based on production data alone did not prove effective in lowering seroprevalence, as was found in other studies [36]. Unfortunately, the culling strategies varied significantly in our study, making statistical comparisons of methods impossible due to sample size. However, the results align with international recommendations that the primary objective of culling programs is to remove animals that are shedding bacteria [37]. Based on this, it seems to be a good practice to base the culling list on ELISA S/P values, as this correlates with bacterial shedding [24]. In addition, it is also helpful to perform faecal PCR as an additional test, and to determine the culling order based on Ct values [38]. Keeping ELISA-positive animals within the herd but not inseminating them can also be an economically feasible strategy, as adult animals can only infect each other under very high infection pressure under practical conditions; moreover, the long incubation period means that they are more likely to be culled before they can start shedding the bacteria [39]. The only exception is cows that shed bacteria in large numbers, which should not be kept in herds under any circumstances because of the significant increase in infection pressure [40]. A summary report published in 2022 found that many national control strategies are based on strict adherence to hygiene and management programs [23], whereas our study shows that more farms (21/29) in Hungary prefer a “test-and-cull” strategy. 

It can be concluded that if a feasible program for a farm can be developed, significant progress seems achievable in controlling the disease over four years. The choice of culling strategy mainly depends on the initial apparent seroprevalence, as in a highly infected farm, where a large number of positive animals makes immediate culling economically challenging. For this purpose, depending on the possibilities of the farm, multiple sampling at the protocol level may be used, or faecal PCR may be implemented as an additional diagnostic tool. In our study, we also showed that farms using faecal PCR as an adjunct achieved a significant reduction in PTB prevalence within four years. In contrast, no significant reduction was detected using complementary ELISA and other culling strategies. The faecal qPCR assay is less widely used due to its high cost, but it is suitable for detecting ‘super-shedders’ and making culling decisions [25].

It is advised to assess the progress of a control program at least every 4 years by determining the apparent seroprevalence. However, it is always essential to choose the same method for this assessment to achieve consistent, and thus comparable data [5]. As a first step, it is necessary for farms to develop feasible methods for calf hygiene and rearing calves free of infection. It is vital to set up a successful control program in the longer term, according to our own experience and as suggested by the international literature [21,23,30,35,41]. The design of “test-and-cull” programs should consider the baseline prevalence and, based on a risk assessment, aim at eliminating “super-shedders” as soon as possible. Repeated ELISA or faecal PCR testing with S/P and Ct values as part of the protocol may be recommended to detect severely infected individuals. For low-infected herds, additional diagnostic testing during the year may also be recommended. If the economic environment allows, immediate culling or exclusion from the breeding of positive animals may be recommended.

## 5. Conclusions

This observational study explored associations between changes in PTB seroprevalence and self-reported levels of awareness and screening, and generated hypotheses to be tested in further studies. Any form of the test-and-cull strategy was shown to be superior to no intervention; furthermore, supplementing culling with newborn calf management that aims at preventing transmission of PTB from adult animals was shown to be superior to merely the test-and-cull strategy. The approach of strengthening culling decisions by performing repeated serological tests was shown to be superior to performing only one test, and additional PCR over repeated serological testing was shown to be superior to repeated serological testing only. These observations could be verified in a study involving a larger number of farms, covering a longer period, and collecting individual data on the culling decisions about each seropositive animal.

## 6. Limitations of the Study

One limitation of the study is its low sample size. We examined 42 farms, which is an adequate number, but at the same time, almost every farm had its own control method, making standardization difficult. There were particular methods used only by one or two farms, making it impossible to perform a statistical analysis of them. There was insufficient statistical power to detect assumed differences between the different levels of added effort in screening and intervention. 

The other important limitation is that we were only able to form an image of the control strategies based on the farm managers’ own reports; thus, a kind of subjectivity cannot be completely ruled out. Another important limitation is that the farm selection process was not random. We needed farms to (1) voluntarily participate in the subsidized testing program, (2) continue testing for at least four years, and (3) if implementing some control program, continue the same measures over the four years. The non-random selection of farms may cause potential bias in the evaluation.

## Figures and Tables

**Figure 1 animals-14-00151-f001:**
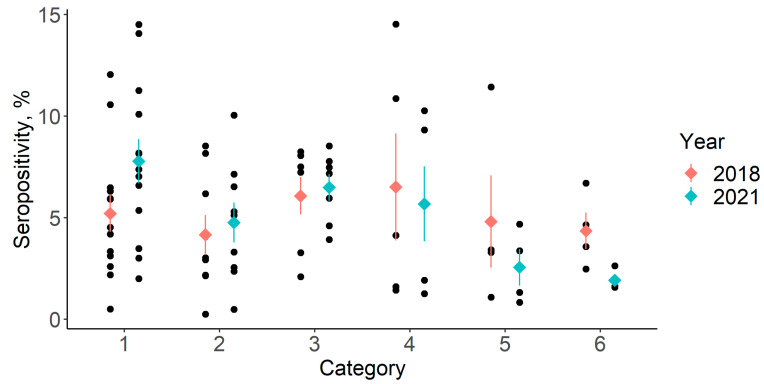
Initial and final seroprevalences. Categories: (1) Annual ELISA + no culling; (2) annual ELISA + culling; (3) annual + complementary individual ELISA + culling; (4) annual +complementary individual ELISA + PCR + culling; (5) annual + complementary individual ELISA + culling + calf management; (6) annual + complementary individual ELISA + PCR + culling + calf management. Dots refer to individual values. Diamonds and error bars indicate the mean and standard error of seroprevalence.

**Table 1 animals-14-00151-t001:** The numbers of farms according to the different aspects of paratuberculosis control.

		Screening				
		Herd-Level ELISA	Herd-Level + Individual ELISA	Herd-Level + Individual ELISA +PCR		
**Intervention**	No intervention	13	0	0	No targeted control	**PTB awareness**
	Planned culling	9	7	5	Some form of control	
	Planned culling and perinatal prevention	0	4	4		

**Table 2 animals-14-00151-t002:** Yearly rates of changes in seropositivity considering different paratuberculosis control strategies.

Item	Answer	Change in Seroprevalence (pp ^1^ per year)	95% CI Lower (pp)	95% CI Upper (pp)	*p*-Value
PTB awareness	yes	−5.1	−11.0	1.0	0.107
	no	12.1	5.0	22.0	<0.001
Individual ELISA	yes	−6.5	−13.0	0.0	0.063
	no	9.6	2.7	18.0	0.004
Individual PCR	yes	−10.9	−20.0	0.0	0.042
	no	5.5	−4.0	12.0	0.081
Planned culling	yes	−4.6	−10.0	1.0	0.150
	no	13.3	6.0	24.0	<0.001
Perinatal prevention	yes	−22.8	−28.0	−12.0	<0.001
	no	7.5	3.0	13.0	<0.001

^1^ pp: percentage point.

**Table 3 animals-14-00151-t003:** The different “test-and-cull” strategies used by farms and the corresponding changes observed in seroprevalence (2021 minus 2018).

Culling Strategy	Number of Farms	Change in Seroprevalence over Four Years (pp ^2^)
ELISA-positive animals not inseminated and culled at the end of lactation	7	−0.4 (min.: −5.1; max. 3.5)
ELISA-positive animals culled based on low milk yield	9	0.5 (min.: −0.1; max −1.2)
Animals with a high ELISA S/P ^1^ value culled sooner	2	1.6 (min.: 1.4; max. 1.9)
PCR-positive animals culled	3	−0.1 (min.: −1.1; max.: 1.4)
Overall	21	0.2 (min: −5.1; max: 3.5)

^1^ A high S/P value was always relative to the herd average. S/P ratio = (optical density (OD) of sample − OD of negative control)/(OD of positive control − OD of negative control) [24]. According to the manufacturer’s instructions, the cut-off values were as follows: negative S/P < 45%; positive: S/P > 55%. ^2^ percentage point.

**Table 4 animals-14-00151-t004:** Initial and final seropositivity values and the percentage point change differences between the two across different levels of PTB control.

Category	1	2	3	4	5	6
2018 (%)	5.2 ± 3.2	4.2 ± 2.9	6.1 ± 2.4	6.5 ± 5.9	4.8 ± 4.6	4.3 ± 1.8
2021 (%)	7.8 ± 3.9	4.8 ± 2.9	6.5 ± 17	5.7 ± 4.1	2.6 ± 1.8 *	1.9 ± 0.5 *
2021–2018 (pp)	2.6 ± 1.5	0.6 ± 1.8	0.4 ± 0.9	−0.8 ± 2.6 *	−2.2 ± 3.2 *	−2.4 ± 1.8 *

Categories: (1) Annual ELISA + no culling; (2) annual ELISA + culling; (3) annual + complementary individual ELISA + culling; (4) annual + complementary individual ELISA + PCR + culling; (5) annual + complementary individual ELISA + culling + calf management; (6) annual + complementary individual ELISA + PCR + culling + calf management. Asterisks show significant differences from reference category 1 (*p* < 0.05).

## Data Availability

The data presented in this study are available on request from the corresponding author.

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
