# Peer review of "Investigating the Prevalence of Paratuberculosis in Hungarian Large-Scale Dairy Herds and the Success of Control Measures over Four Years"

_animals, 2024, doi:10.3390/ani14010151_

Round 1
Reviewer 1 Report
Comments and Suggestions for Authors
General comments
It is understood that the authors focused on comparison of ELISA seroprevalence of PTB over four years in three main categories of dairy farms:
- No control measures
- Test and cull measures
- Test ad cull, plus immediate or early (?) separation of calves from dams
It seems that statistical analysis (based on questionnaire ?) confirmed earlier knowledge and expectations:
- increase of seroprevalence in category 1.
- Slight or no reduction of seroprevalence in category 2.
- Significant reduction in seroprevalence in category 3.
Although these results can not be regarded as scientifically new, they confirm the present general concept about reduction of PTB in dairy herds and they contain some useful anecdotal information. However the manuscript has several deficiencies which make the impression that it is a bit premature and need much more precision and care for further improvements (see some examples below). The authors should try to describe their Methods and Results, with more scientific precision and the Discussion sections also much more clearly, and try to avoid repetitious statements.
Specific comments
- Title is incorrect: no “eradication” of PTB can be mentioned in these studies.
- Abstract needs clarity and details about the number of farms (and possibly of the total number of animals in 4 years) included in each of the above three categories. Possibly a uniform questionnaire was used, for the farmers, which should also be mentioned.
- It is necessary to mention “Commercial ELISA” and define the PCR used.
- Also in Abstract: one sentence describe the “Aims” (L25-26), he next sentence describes a different “Objective” (L26-27).
- Methods are poorly described: description of subgroups within the above categories is not clear (L:115-140). t
- There would be a need for a more simple, more straightforward and more consequent description of the categories (or: groups) once they deserve independent statistical analysis.
- Fecal PCR tests are important part of the studies but description of the PCR and qPCR methods are missing in the Methods section.
- Regarding the qPCR, the definition of “high shedder” should also be given
- Table 1. should indicate the number of farms (and possibly of the total number of animals in 4 years) for each subgroups in order to appreciate the values of statistical analysis.
- Table 1. Heading is not clear (what means: “considering the years of sampling” and why is it needed to be there ?)
- Table 1. Footnote: the comments No. 2 and No3. are both the same text, although No3 should refer to the PCR (not again to the ELISA).
- The frequent mentioning and no exact definitions of different “management strategies” is disturbing and makes the basis of analyses. Please clarify the different management strategies for each cases once they were different (i.e. complete and immediate separation of newborn calves from the seropositive dams ? or giving colostrum for the dam (for how long?)
- Conclusion: should say “here we confirm” (instead of we can draw the conclusion) (L:272).
Finally:
It is difficult to see why this manuscript would fit to the journal "Animals" within the MDPI. To my best judgement it would rather fit to Veterinary Sciences.
It is not really dealing with the animals, but it deals with veterinary epidemiology and herd management.
Comments on the Quality of English LanguageEnglish is adeqate.
Author Response
Reviewer 1
General comments
It is understood that the authors focused on comparison of ELISA seroprevalence of PTB over four years in three main categories of dairy farms:
No control measures
Test and cull measures
Test ad cull, plus immediate or early (?) separation of calves from dams
It seems that statistical analysis (based on questionnaire ?) confirmed earlier knowledge and expectations:
increase of seroprevalence in category 1.
Slight or no reduction of seroprevalence in category 2.
Significant reduction in seroprevalence in category 3.
Although these results can not be regarded as scientifically new, they confirm the present general concept about reduction of PTB in dairy herds and they contain some useful anecdotal information. However the manuscript has several deficiencies which make the impression that it is a bit premature and need much more precision and care for further improvements (see some examples below). The authors should try to describe their Methods and Results, with more scientific precision and the Discussion sections also much more clearly, and try to avoid repetitious statements.
AU: Thank you for your effort to read our manuscript, and for your valuable comments and suggestions. We tried to do our best to improve the overall quality of the manuscript. Since the reviewers had many suggestions, we did not use the track changes function, instead, the changed parts are indicated in yellow.
Specific comments
Rev.: Title is incorrect: no “eradication” of PTB can be mentioned in these studies.
AU: Indeed, there was no eradication, just some (sometimes unsuccessful) efforts to eradicate the disease, therefore we changed the title by using the word „control” instead of „eradication”. Additionally, we stopped using the term „eradication” throughout the text in the context of our study.
Rev.: Abstract needs clarity and details about the number of farms (and possibly of the total number of animals in 4 years) included in each of the above three categories. Possibly a uniform questionnaire was used, for the farmers, which should also be mentioned.
AU: The number of farms (42) and the animals (on average, 32009 yearly) are already in the text. We supplemented the abstract with the total number of animals during the four years 127372). Yes, one time every year of the study, a uniform list of questions was put to the vets or farm managers to gain data about the PTB control protocol of the farm. We added this information to Materials and Methods.
Rev: It is necessary to mention “Commercial ELISA” and define the PCR used.
AU: We mentioned „commercial ELISA” in the abstract, and also its name, since another reviewer requested it. PCR is not a very important part of the study since it was the farm's decision to do additional PCR. This was just additional information about their PTB control protocol and was not an experiment. Therefore, it is not mentioned in the abstract. Please see also our answer to your later comment related to the PCR.
Rev: Also in Abstract: one sentence describe the “Aims” (L25-26), he next sentence describes a different “Objective” (L26-27).
AU: We deleted the word „objective”, and we combined the two sentences.
Rev.: Methods are poorly described: description of subgroups within the above categories is not clear (L:115-140). There would be a need for a more simple, more straightforward and more consequent description of the categories (or: groups) once they deserve independent statistical analysis.
AU: We tried to rephrase and clarify the text.
Rev: Fecal PCR tests are important part of the studies but description of the PCR and qPCR methods are missing in the Methods section. Regarding the qPCR, the definition of “high shedder” should also be given
AU: PCR was not part of our study, this is just a piece of information the farm managers gave us about their PTB control protocols. It was not planned by us, and just a few farms used it. Otherwise, the lab used the Adiavet ParaTB Real-Time kit (Bio-X Diagnostics S.A., Rochefort, Belgium) for the qPCR run. The high shedders were selected based on their Ct values, but there was no uniform threshold. The farm managers decided the threshold for culling based on the farm average.
Rev.: Table 1. should indicate the number of farms (and possibly of the total number of animals in 4 years) for each subgroups in order to appreciate the values of statistical analysis.
Table 1. Heading is not clear (what means: “considering the years of sampling” and why is it needed to be there ?)
Table 1. Footnote: the comments No. 2 and No3. are both the same text, although No3 should refer to the PCR (not again to the ELISA).
AU: We inserted a new table about the different aspects of paratuberculosis control (new Table 1), that contains the number of the farms in each category.
Rev.: The frequent mentioning and no exact definitions of different “management strategies” is disturbing and makes the basis of analyses. Please clarify the different management strategies for each cases once they were different (i.e. complete and immediate separation of newborn calves from the seropositive dams ? or giving colostrum for the dam (for how long?)
AU: We call management strategies all the important actions which are not test-and-cull, but rather calf management and hygiene, including immediate separation of calf from the dam, giving MAP negative colostrum for the newborns, etc. as it is written in our previous paper (Vass-Bognár et al., Prev Vet Med, 2022 (https://doi.org/10.1016/j.prevetmed.2022.105719). We treated all these actions as one: management procedures, and we did not differentiate them. We tried to clarify the text accordingly.
Rev.: Conclusion: should say “here we confirm” (instead of we can draw the conclusion) (L:272).
AU: The text is changed.
Finally:
It is difficult to see why this manuscript would fit to the journal "Animals" within the MDPI. To my best judgement it would rather fit to Veterinary Sciences. It is not really dealing with the animals, but it deals with veterinary epidemiology and herd management.
AU: Thank you for your suggestion. We felt the topic fits Animals, but we can change our intentions if the editors say so.
Reviewer 2 Report
Comments and Suggestions for Authors
The authors investigated with a retrospective study the prevalence of paratuberculosis for 4 consecutive years (2018-2021) in 42 large-scale dairy herds in Hungary.
The aim was to monitor the progress of the eradication programs implemented by Hungarian farms compared to farms without paratuberculosis control measures. Overall, in 15/42 farms that did not use control measures the seroprevalence increased by 12.1% in four years, while in the farms with control strategies (any) the seroprevalence was reduced by an average of at least 5.1%.
In farms that implemented both the test and cull strategy and managerial strategies, the reduction in seroprevalence reached 22.8% over the four years, concluding that test and cull alone is not sufficient for a PTB eradication program, but is also necessary good hygiene management of the calf at birth and possibly also use diagnostic tests (ELISA-PTB or qPCR on faeces) to identify and eliminate infected animals.
The research is structured correctly, the results are clear and the discussion is well articulated. The limitations of the study (research carried out in 4 years instead of 5 due to the cessation of state funding) relating to the number of companies tested (which in any case represent 15% of the Hungarian dairy cattle herd) are also clearly expressed.
Suggestions:
Abstract:
- line 28: between, not be-tween
- lines 32-37: I would reverse the sentences by putting first "The farms with control strategy had the 34 chance to reduce MAP seroprevalence on average 5.1%. For the farms that carried out a complex 35 program with both “test-and-cull” and management strategies, there was a proportional reduction 36 in apparent seroprevalence of an average of 22.8% over the four years." and then "Fifteen of the 32 sampled farms had no measures in place to control paratuberculosis. On these farms, the seroprev-33 alence increased with an average 12.1% over the four years."
Introduction:
- line 56: there is a missing mention to the WOAH list of animal diseases. Furthermore, if I could suggest, it is necessary to mention the "Commission Implementing Regulation (EU) 2018/1882 of 3 December 2018 on the application of certain disease prevention and control rules to categories of listed diseases and establishing a list of species and groups of species posing a considerable risk for the spread of those listed diseases" which include Paratuberculosis in Bison ssp., Bos ssp., Bubalus ssp., Ovis ssp., Capra ssp., Ca melidae, Cervidae as category E of listed diseases. On these diseases there is a need for surveillance within the Union, as referred to in Article 9(1)(e) of Regulation (EU) 2016/429.
Materials and Methods:
- line 102: it is not mentioned what was the minimum number of animals per farm to be included in the study.
- line 140: Were farms that were not tested for disease incidence or that stopped the eradication program during the testing period also excluded from the total number of farms and animals tested? it is not clear.
Results, discussion and conclusions: nothing to correct or suggest.
Author Response
Reviewer 2
Comments and Suggestions for Authors
The authors investigated with a retrospective study the prevalence of paratuberculosis for 4 consecutive years (2018-2021) in 42 large-scale dairy herds in Hungary.
The aim was to monitor the progress of the eradication programs implemented by Hungarian farms compared to farms without paratuberculosis control measures. Overall, in 15/42 farms that did not use control measures the seroprevalence increased by 12.1% in four years, while in the farms with control strategies (any) the seroprevalence was reduced by an average of at least 5.1%.
In farms that implemented both the test and cull strategy and managerial strategies, the reduction in seroprevalence reached 22.8% over the four years, concluding that test and cull alone is not sufficient for a PTB eradication program, but is also necessary good hygiene management of the calf at birth and possibly also use diagnostic tests (ELISA-PTB or qPCR on faeces) to identify and eliminate infected animals.
The research is structured correctly, the results are clear and the discussion is well articulated. The limitations of the study (research carried out in 4 years instead of 5 due to the cessation of state funding) relating to the number of companies tested (which in any case represent 15% of the Hungarian dairy cattle herd) are also clearly expressed.
AU: Thank you for your valuable comments. We tried to do our best to clarify the manuscript. Since the reviewers had many suggestions, we did not use the track changes function, instead, the changed parts are indicated in yellow.
Suggestions:
Abstract:
Rev.: - line 28: between, not be-tween
AU: Thank you, it is corrected.
Rev.: lines 32-37: I would reverse the sentences by putting first "The farms with control strategy had the 34 chance to reduce MAP seroprevalence on average 5.1%. For the farms that carried out a complex 35 program with both “test-and-cull” and management strategies, there was a proportional reduction 36 in apparent seroprevalence of an average of 22.8% over the four years." and then "Fifteen of the 32 sampled farms had no measures in place to control paratuberculosis. On these farms, the seroprev-33 alence increased with an average 12.1% over the four years."
AU: Thank you, it is corrected now.
Introduction:
Rev.: line 56: there is a missing mention to the WOAH list of animal diseases. Furthermore, if I could suggest, it is necessary to mention the "Commission Implementing Regulation (EU) 2018/1882 of 3 December 2018 on the application of certain disease prevention and control rules to categories of listed diseases and establishing a list of species and groups of species posing a considerable risk for the spread of those listed diseases" which include Paratuberculosis in Bison ssp., Bos ssp., Bubalus ssp., Ovis ssp., Capra ssp., Camelidae, Cervidae as category E of listed diseases. On these diseases there is a need for surveillance within the Union, as referred to in Article 9(1)(e) of Regulation (EU) 2016/429.
AU: Thank you for your suggestion. We cited the WOAH list and also the European regulations.
Materials and Methods:
Rev.: line 102: it is not mentioned what was the minimum number of animals per farm to be included in the study.
AU: We included large-scale farms in the study, a minimum of 200 dairy cows per farm.
Rev.: line 140: Were farms that were not tested for disease incidence or that stopped the eradication program during the testing period also excluded from the total number of farms and animals tested? it is not clear.
AU: Yes, the farms not tested for disease incidence for all 4 years or stopped the control program during the four-year study period were excluded from the analysis.
Rev.: Results, discussion and conclusions: nothing to correct or suggest.
Reviewer 3 Report
Comments and Suggestions for Authors
The study, entitled “Investigating the prevalence of paratuberculosis in Hungarian large-scale dairy herds and the success of eradication programs over four years” aimed to retrospectively assess the prevalence of paratuberculosis (PTB) in large-scale dairy herds in Hungary between January 2018 and December 2021. The primary objective was to monitor the effectiveness of PTB eradication programs in these herds and observe those without any measures to control paratuberculosis. The significance of the study lies in its contribution to understanding effective PTB control strategies. By highlighting the limitations of relying solely on a "test-and-cull" approach and emphasizing the importance of hygiene management measures during calving and newborn care, the research provides valuable insights for the dairy industry. The findings contribute to the ongoing efforts to mitigate the economic losses associated with PTB and enhance the success of eradication programs in large-scale dairy herds.
In general, the manuscript is well-organized, well-written, and the topic is generally intriguing and potentially valuable for readers. However, there are a few typographical and grammatical errors that require correction before the manuscript can be considered for publication.
I have few minor queries:
> Incorporate the name of the test used into the Abstract section.
> Line No. 35: Use complete name for MAP when referencing it for the first time.
> Line No. 40: Add a keyword “seroprevalence” before dairy cattle.
> Line No. 55: Listed as what? I mean, is it listed under category A, B, or C of the disease by WHO?
> Line No. 66 and Line 204-205 (Table 2): Replace commas with dots (.) between numbers, where applicable.
> Line No. 101: Organize the Materials and Methods section into distinct subsections, such as description of study area, estimation of sample size, sample collection etc.
> Line No. 142-49: Introduce a section heading titled "Statistical Analysis" for this paragraph.
Comments on the Quality of English Language
Minor editing of English language required
Author Response
Reviewer 3.
Comments and Suggestions for Authors
The study, entitled “Investigating the prevalence of paratuberculosis in Hungarian large-scale dairy herds and the success of eradication programs over four years” aimed to retrospectively assess the prevalence of paratuberculosis (PTB) in large-scale dairy herds in Hungary between January 2018 and December 2021. The primary objective was to monitor the effectiveness of PTB eradication programs in these herds and observe those without any measures to control paratuberculosis. The significance of the study lies in its contribution to understanding effective PTB control strategies. By highlighting the limitations of relying solely on a "test-and-cull" approach and emphasizing the importance of hygiene management measures during calving and newborn care, the research provides valuable insights for the dairy industry. The findings contribute to the ongoing efforts to mitigate the economic losses associated with PTB and enhance the success of eradication programs in large-scale dairy herds.
In general, the manuscript is well-organized, well-written, and the topic is generally intriguing and potentially valuable for readers. However, there are a few typographical and grammatical errors that require correction before the manuscript can be considered for publication.
AU: Thank you for reading and reviewing our manuscript and for your valuable comments and suggestions. Since the reviewers had many suggestions, we did not use the track changes function, instead, the changed parts are indicated in yellow.
I have few minor queries:
Rev.: Incorporate the name of the test used into the Abstract section.
AU: It is now included in the text.
Rev.: Line No. 35: Use complete name for MAP when referencing it for the first time.
AU: We decided to mention it even earlier, and its name is written completely.
Rev.: Line No. 40: Add a keyword “seroprevalence” before dairy cattle.
AU: Thank you, it is added now. We also added the keyword ’biosecurity’.
Rev.: Line No. 55: Listed as what? I mean, is it listed under category A, B, or C of the disease by WHO?
AU: Paratuberculosis is a ’category E’ disease. The other reviewers also requested a reference for the WOAH list, so we included it.
Rev.: Line No. 66 and Line 204-205 (Table 2): Replace commas with dots (.) between numbers, where applicable.
AU: Thank you, it is corrected.
Rev.: Line No. 101: Organize the Materials and Methods section into distinct subsections, such as description of study area, estimation of sample size, sample collection etc.
Line No. 142-49: Introduce a section heading titled "Statistical Analysis" for this paragraph.
AU: The Materials and Methods section is now divided into subsections.
Reviewer 4 Report
Comments and Suggestions for Authors
This is a flawed study that in its present form is not suitable for publication. The methodology around assessing the level of compliance is obscure and what is presented for test and cull as a method of control indicates that this category was not a unified approach and simply not adequately nor uniformly adhered to allow the comparisons claimed.
Simple summary
Lines 19-20: “we found that farms that base their strategy only on blood testing and culling infected animals cannot sufficiently reduce the incidence of the disease.” By the authors’ own admission the study was not long enough to assess this. In addition, the test and cull programme took too many different forms on the farms that used this strategy to allow a collective view of the success or otherwise and there was no assessment of the degree of compliance in the test and cull or of any other of the examined control strategies.
Abstract
Line 25: Mentions retrospectively in relation to investigating prevalence of Ptb on the study farms. This is also mentioned in line 93 in the introduction, but not subsequently.
Lines 32-34: state that the seroprevalence on the 15 farms without measures to control paratuberculosis increased with an average of 12.1% over the four years. In the results, line 155-160, again the average increase of nearly 12% is quoted, however, it also states that in six of the (15) farms the average increase in seroprevalence was 3.1% and in the remaining 9 farms the average increase in seroprevalence was 1.8%. These data are difficult to reconcile without more explanation.
Lines 36 to 37: Gives a proportional reduction in apparent seroprevalence of an average of 22.8%. This is misleading, as while it may be correct, this way of describing the data is not used in any other category and the average prevalence at the end of the four years for this category was 2%, having been 4.5% (as judged from the trend line in figure 1).
Introduction
Line 46: “mostly affects dairy cows” is misleading. The disease can cause severe problems in beef cow herds, sheep flocks and goat herds.
Line 57: This statement requires to be referenced.
Line 64: This reference does not include one mention of seroprevalence.
Lines 94): “large-scale dairy herds” must be defined and the rationale for this focus clarified.
Materials and methods
One of the important design aspects of this study is that according to the introduction and the abstract this was done retrospectively. This word does not appear in the material and methods. It is important to detail when the study was carried out, i.e. when were the farm managers interviewed.
Line 106: selection of farms for study appears to be non-random and therefore subject to potential source of biases that are not acknowledged.
Line 109: Seroprevalence is qualified by “apparent”, but there is no definition of this. Presumably it is the proportion of animals that tested positive for antibody in the ELISA on the day of the annual herd test. There doesn’t seem to be anything apparent about this.
Line 108: mentions the commercial laboratory where all test results for study was done. This paragraph could be turned around to put the test used first; followed by the restriction to one test per year used to assess progress and that all results used were from tests carried out in one commercial laboratory. No mention is made of how the samples were collected and by whom nor whether the control programmes were designed by the farm's veterinarian or otherwise.
Line 120: The word “monitor” is used in relation to contact made by telephone or in person. It is not stated when this was carried out and the use of the word conflicts with the statement that the study was retrospective as monitor indicates on-going activity.
Lines 122-127: states that farms were categorised in to two main groups, i.e. seroprevalence <5% and seroprevalence >=5%. It then goes on to mention farms with a control strategy in place. This is confusing and needs to be rewritten to clarify. (In figure 1 the trend lines are the two non control (<5% and >=5%) and the two groups applying a control programme.
Line 125: Uses the term “rate of PTB seroprevalence over the study period.” Again difficult to understand. Is it simply the change in seroprevalence over the study period?
Line 128: “We evaluated the impact of the diagnostic methods used to make the culling decision, i.e. which method was likely to have achieved a more significant reduction in seroprevalence in the farms using the “test and cull” strategy.” This sentence is difficult to understand.
Lines 130-132: “The programme was considered complete if it was continuously followed over the four years and no substantial modification was made.” No explanation is then given as to whether incomplete farms were excluded from the analysis.
Lines 134-141: This describes the hygiene and management measures. These are low level interventions. No mention is made of colostrum and the control programme only appears to apply to animals that are known to be test positive, i.e. not feeding raw milk from test positive mothers. There is no standard questionnaire provided and no independent assessment of the degree of on-farm compliance with the control programme or with culling. The latter is important as at one point it says that test positive animals are culled immediately, but then says test positive animals are calved separately and it is clear that culling positives depended on several factors and did not necessarily result in the test positive animal being removed from the herd.
Line 140: “farms not tested for disease incidence” There is no mention of disease incidence elsewhere and presumably this should have “farms not tested for seroprevalence”
Line 144-145: Mentions “infected cows” as being a dependent variable in the model. Again, I assume this should have been “seropositive cows”.
Results
Line 151-153: This paragraph is difficult to understand and the data would be better presented in a table for the different categories identified in the material and methods.
Lines 155-163 This is another confusing passage that would benefit from a table. It states that the average increase in apparent seroprevalence in 6 farms was 3.1% and the remaining nine farms the average increase in seroprevalence was 1.8% over the four-year period. However, confusingly the Poisson regression model found no statistical difference between these groups “with an average increase of nearly 12%.” In table 1 it gives the change in seroprevalence of 12.1% for this group.
Line 165-167: States that there were no differences in initial apparent seroprevalence between the groups. How this was examined is not made clear and figure 1 indicates a starting prevalence for the no-control < 5% of around 2.5% and around 5.5% for test and cull only. While the 5% and over no control has a starting prevalence of almost 8%. The arbitrary splitting of no control into high and low prevalence has dubious value unless that same split is made for the other categories.
Lines 187-190: Here, for the first time, the proportional reduction in “apparent” seroprevalence is used to show the benefit of test and cull plus management strategies. This is only valid if the same was done for the other trend lines otherwise it serves to exaggerate any change in the test and cull plus management.
Lines 191-192: “test and cull” strategy made virtually no progress…..with an average increase in seroprevalence of 0.1%..” However, figure 1 shows the trend line to fall from 5.5% to 5%.
Table 1: This is labelled “Changes in the rate of infection considering the years of sampling and the different strategies”. This is a meaningless title. Presumably it should be “Changes in the seroprevalence over the four-year period of study” or similar. The table should include the number of farms for each category.
Figure 1: Is labelled changes in the apparent seroprevalence …. over a four-year period and has trend lines connecting the point prevalence values for each year. Different categories are used for the four trend lines than in table 1, but the maximum seroprevalence is only 11% for the no-control >=5%. That is these data are described similarly to those in table 1, but don’t relate. The use of error bars would help in visualising these distributions.
Table 2: This table shows the different test and cull strategies. None of the categories are based on removing all ELISA positives. Almost all require some other test to be done. This does not represent a uniform test and cull strategy that can be assessed.
Discussion
Line 210-211: typo to be corrected.
Line 212-215: Claims that the increase (in seroprevalence) found in the current study is likely to be lower than the actual Hungarian average as 27 of the farms had some sort of eradication programme for Ptb in place. This claim cannot be made as it is not stated how the study population differs from the national population in this respect and previously it claims that paratuberculosis is a greater problem in large herds and this study focused on large herds and accounted for “15% of the cattle population”
Lines 220-222: this statement cannot be supported by the findings as there was no evaluation of the adherence to the management aspect of the control programme and no common or rigorous approach to test and cull and therefore this study has not examined the effectiveness of either approach.
Lines 224:226: There is a claim that culling on production alone was the least effective strategy (and this is not a test and cull strategy), but there is no analysis to support this and there was only one of the study farms that took this approach.
Lines 230-232: The approach of culling based on S/P value can be questioned and a more extensive evaluation of the recent literature on the subject is advised.
Lines 242-245: Here it states that “Compliance with management measures over the four-year study period has been unsuccessful in most cases.” This is problematic, because no information to support the degree of compliance has been reported in the study and secondly if compliance was generally poor (and it would appear to have been as judged by the variation in approach to test and cull) then it renders any comparison between the different categories meaningless.
Lines 249-250: “”makes immediate culling impossible”. This is not correct. Culling 5% of cows that are affected by a production limiting disease is unlikely to lead to financial ruin. It is certainly to be expected that herd managers might make such a claim to retain cows and in the short term that might result in higher milk sales. Such a claim cannot be made without some supporting evidence or argument.
Lines 252-257: Claims that using faecal PCR…. achieved a significant reduction of the Ptb prevalence within four years. No statistical analysis is presented to support this. There were seven farms that used PCR in various ways and in only one farm where PCR positives were culled after calving was the change in prevalence (-5%) different from other farms in the test and cull categories.
Conclusions
Lines 272-276: These conclusions may be arrived by reviewing the literature, but the work done in this study does not allow them to be reached.
Limitations of the study
The limitations of the study are the lack of randomness in farm selection without any acknowledgement of the potential biases; and the failure to properly assess the degree of compliance to the chosen programmes at the farm level; or to assess the degree of success of the management programmes in achieving the objective of reducing the opportunity for the transmission of the disease.
Comments on the Quality of English Language
The quality of the English is good, but there is some confusion present in the use of seroprevalence, rates, infection.
Author Response
Reviewer 4.
Comments and Suggestions for Authors
Rev.: This is a flawed study that in its present form is not suitable for publication. The methodology around assessing the level of compliance is obscure and what is presented for test and cull as a method of control indicates that this category was not a unified approach and simply not adequately nor uniformly adhered to allow the comparisons claimed.
AU: Thank you for reading and reviewing our manuscript and for your valuable comments. Reading our manuscript after the reviews, it became obvious that the text is unclear in several places. We tried to do our best to make it clearer. Since we received comments and suggestions from five reviewers, we did not use the track changes function of Word. Instead, the changed text parts are indicated in yellow.
Simple summary
Rev.: Lines 19-20: “we found that farms that base their strategy only on blood testing and culling infected animals cannot sufficiently reduce the incidence of the disease.” By the authors’ own admission the study was not long enough to assess this. In addition, the test and cull programme took too many different forms on the farms that used this strategy to allow a collective view of the success or otherwise and there was no assessment of the degree of compliance in the test and cull or of any other of the examined control strategies.
AU: Indeed, according to the literature, it is advised to judge the success of most PTB control programs after five years (Nielsen and Toft, JDS, 2011. https://doi.org/10.3168/jds.2010-3817). However, there are other, more recent studies (e.g. Scarpellini et al., Prev Vet Med, 2023. https://doi.org/10.1016/j.prevetmed.2023.105923) evaluating the success of the programs after four years. We believe 4 years should be enough for the evaluation, or at least to see the main tendencies. And yes, we agree that the test and cull programme took too many different forms on the farms. Our study was not an experiment or a national PTB control program driven from above with a uniform strategy. The farms decided on their own strategy, we just collected their data. We believe that it has a meaning if we find significant differences despite the high variability of the strategies.
Abstract
Rev.: Line 25: Mentions retrospectively in relation to investigating prevalence of Ptb on the study farms. This is also mentioned in line 93 in the introduction, but not subsequently.
AU: The word retrospectively was deleted from this sentence.
Rev.: Lines 32-34: state that the seroprevalence on the 15 farms without measures to control paratuberculosis increased with an average of 12.1% over the four years. In the results, line 155-160, again the average increase of nearly 12% is quoted, however, it also states that in six of the (15) farms the average increase in seroprevalence was 3.1% and in the remaining 9 farms the average increase in seroprevalence was 1.8%. These data are difficult to reconcile without more explanation.
AU: Thank you for your comment, it is misleading, indeed. The 12% is a growth rate per year. The abstract was rephrased.
Rev.: Lines 36 to 37: Gives a proportional reduction in apparent seroprevalence of an average of 22.8%. This is misleading, as while it may be correct, this way of describing the data is not used in any other category and the average prevalence at the end of the four years for this category was 2%, having been 4.5% (as judged from the trend line in figure 1).
AU: Yes, thank you. We corrected the text and inserted a new Figure. We also distinguished between percentage and percentage points, meaning a rate of change or an amount of change, respectively.
Introduction
Rev.: Line 46: “mostly affects dairy cows” is misleading. The disease can cause severe problems in beef cow herds, sheep flocks and goat herds.
AU: We agree. Dairy cattle are our focus, perhaps this is why we feel the problems in the dairy industry are more numerous. Anyway, we rephrased the text.
Rev.: Line 57: This statement requires to be referenced.
AU: We inserted a reference in the text.
Rev.: Line 64: This reference does not include one mention of seroprevalence.
AU: Yes, indeed, they used milk serology data for their analysis. We included this in the text.
Rev.: Lines 94): “large-scale dairy herds” must be defined and the rationale for this focus clarified.
AU: The average farm size in Hungary is 500 cows, the number of large farms far exceeds that of small farms (smaller than 200 cows). We tried to show the picture of average size Hungarian farms, therefore we asked larger ones to contribute to the study. The farm size is defined in the Materials and Methods section. Nevertheless, we deleted „large-scale” from this part.
Materials and methods
Rev.: One of the important design aspects of this study is that according to the introduction and the abstract this was done retrospectively. This word does not appear in the material and methods. It is important to detail when the study was carried out, i.e. when were the farm managers interviewed.
AU: Thank you for your comment. The term „retrospectively” is misleading here. The farms did one sampling per year, this was the basis of the examination. Retrospective means that we collected all sampling data from the four years and performed the analysis after the data collection. Farm managers were asked every year after the sampling about their PTB control programs (what they did, whether was there any change, etc). These data were used for the possible exclusion of farms, and the analysis was done after collecting all data from the four years. We tried to clarify this in the Material and Methods section.
Rev: Line 106: selection of farms for study appears to be non-random and therefore subject to potential source of biases that are not acknowledged.
AU: Yes, we agree, selection can not be random. We needed farms that 1) voluntarily participate in the subsidized testing program, 2) continue testing for at least four years, 3) if implementing some control program, continue the same measures over the four years. We are happy to find 42 farms that can be included in this study. We tried to acknowledge this in the Limitations section.
Rev.: Line 109: Seroprevalence is qualified by “apparent”, but there is no definition of this. Presumably it is the proportion of animals that tested positive for antibody in the ELISA on the day of the annual herd test. There doesn’t seem to be anything apparent about this.
AU: Serologic tests used for conducting seroepidemiologic and prevalence studies are typically imperfect and produce false-positive and false-negative results. This is why the seropositive rate (apparent prevalence) does not typically reflect the true prevalence of the disease or condition of interest. Not all subjects with positive tests are diseased, and not all with negative tests are disease-free. This is why the prevalence derived from these studies, the so-called “apparent prevalence” (pr), is not necessarily an unbiased estimation of the true prevalence (π), the true proportion of diseased people in the population or the study sample. The pr (the apparent prevalence) is defined as the portion of tested people with a positive test (T +). Therefore:
pr = P (T+) = TPR + FPR
where TPR and FPR are true-positive and false-positive rates. We did not perform a true prevalence estimation in our study, therefore we used the term apparent prevalence. We included this example in the text.
Rev.: Line 108: mentions the commercial laboratory where all test results for study was done. This paragraph could be turned around to put the test used first; followed by the restriction to one test per year used to assess progress and that all results used were from tests carried out in one commercial laboratory. No mention is made of how the samples were collected and by whom nor whether the control programmes were designed by the farm's veterinarian or otherwise.
AU: The paragraph is now turned around, starting with the sampling and the test description. The samples were taken by the farm veterinarians. The control programs were designed by the farm vets, we described this in the text.
Rev.: Line 120: The word “monitor” is used in relation to contact made by telephone or in person. It is not stated when this was carried out and the use of the word conflicts with the statement that the study was retrospective as monitor indicates on-going activity.
AU: Please see our explanation for your first comment in the Materials section.
Rev.: Lines 122-127: states that farms were categorised in to two main groups, i.e. seroprevalence <5% and seroprevalence >=5%. It then goes on to mention farms with a control strategy in place. This is confusing and needs to be rewritten to clarify. (In figure 1 the trend lines are the two non control (<5% and >=5%) and the two groups applying a control programme.
AU: After careful consideration, we decided to delete this part.
Rev.: Line 125: Uses the term “rate of PTB seroprevalence over the study period.” Again difficult to understand. Is it simply the change in seroprevalence over the study period?
AU: The text is corrected.
Rev.: Line 128: “We evaluated the impact of the diagnostic methods used to make the culling decision, i.e. which method was likely to have achieved a more significant reduction in seroprevalence in the farms using the “test and cull” strategy.” This sentence is difficult to understand.
AU: The Mat&Met section was rephrased.
Rev.: Lines 130-132: “The programme was considered complete if it was continuously followed over the four years and no substantial modification was made.” No explanation is then given as to whether incomplete farms were excluded from the analysis.
AU: Yes, they were excluded. It was mentioned in the Material and Methods, but now the text is restructured and hopefully clearer.
Rev.: Lines 134-141: This describes the hygiene and management measures. These are low level interventions. No mention is made of colostrum and the control programme only appears to apply to animals that are known to be test positive, i.e. not feeding raw milk from test positive mothers. There is no standard questionnaire provided and no independent assessment of the degree of on-farm compliance with the control programme or with culling. The latter is important as at one point it says that test positive animals are culled immediately, but then says test positive animals are calved separately and it is clear that culling positives depended on several factors and did not necessarily result in the test positive animal being removed from the herd.
AU: We mentioned the colostrum supply, as this is also an important management measure against the spread of PTB. We have found that the management (or biosecurity) measures are important (see Vass-Bognár et al., Prev Vet Med, 2022 https://doi.org/10.1016/j.prevetmed.2022.105719) and can be high-level interventions, as found in our study, and in others (Scarpellini et al., Prev Vet Med, 2023. https://doi.org/10.1016/j.prevetmed.2023.105923).
We want to highlight here that there was no standard control program (developed e.g. us or a ’National Body’). The farm managers decided, with the help of the farm vet, about their own farm-tailored control program. It is hard to assess the degree of compliance without a standard protocol. The culling decision also differed farm by farm, as shown in Table 3. Some used production levels, some performed one or two extra ELISA besides the yearly serology, and some used PCR. If they used ELISA, some decided to cull the positive animals immediately, some decided not to inseminate them and cull them at the end of lactation. It was impossible to perform statistical analysis for all of them due to the low sample sizes. When performing the statistics, we considered all the farms doing some control as a „farm with control program”, and all forms of control programs were considered as a „control program”.
Rev.: Line 140: “farms not tested for disease incidence” There is no mention of disease incidence elsewhere and presumably this should have “farms not tested for seroprevalence”
AU: Yes, it is seroprevalence. The text is corrected.
Rev.: Line 144-145: Mentions “infected cows” as being a dependent variable in the model. Again, I assume this should have been “seropositive cows”.
AU: Yes, indeed. The text is corrected.
Results
Rev.: Line 151-153: This paragraph is difficult to understand and the data would be better presented in a table for the different categories identified in the material and methods.
AU: A new table is now included.
Rev.: Lines 155-163 This is another confusing passage that would benefit from a table. It states that the average increase in apparent seroprevalence in 6 farms was 3.1% and the remaining nine farms the average increase in seroprevalence was 1.8% over the four-year period. However, confusingly the Poisson regression model found no statistical difference between these groups “with an average increase of nearly 12%.” In table 1 it gives the change in seroprevalence of 12.1% for this group.
AU: We rephrased the text.
Rev.: Line 165-167: States that there were no differences in initial apparent seroprevalence between the groups. How this was examined is not made clear and figure 1 indicates a starting prevalence for the no-control < 5% of around 2.5% and around 5.5% for test and cull only. While the 5% and over no control has a starting prevalence of almost 8%. The arbitrary splitting of no control into high and
AU: The total Results section has been rephrased for clarity.
Rev.: Lines 187-190: Here, for the first time, the proportional reduction in “apparent” seroprevalence is used to show the benefit of test and cull plus management strategies. This is only valid if the same was done for the other trend lines otherwise it serves to exaggerate any change in the test and cull plus management.
AU: The total Results section has been rephrased for clarity.
Rev.: Lines 191-192: “test and cull” strategy made virtually no progress…..with an average increase in seroprevalence of 0.1%..” However, figure 1 shows the trend line to fall from 5.5% to 5%.
AU: Thank you for your comment. We deleted the Figure 1, and a new Figure was made.
Rev.: Table 1: This is labelled “Changes in the rate of infection considering the years of sampling and the different strategies”. This is a meaningless title. Presumably it should be “Changes in the seroprevalence over the four-year period of study” or similar. The table should include the number of farms for each category.
AU: Yes, it is seroprevalence. The table heading is corrected. Also, the number of farms in each category is displayed, as another reviewer asked. It is now Table 2.
Rev.: Figure 1: Is labelled changes in the apparent seroprevalence …. over a four-year period and has trend lines connecting the point prevalence values for each year. Different categories are used for the four trend lines than in table 1, but the maximum seroprevalence is only 11% for the no-control >=5%. That is these data are described similarly to those in table 1, but don’t relate. The use of error bars would help in visualising these distributions.
AU: Indeed, they are not related. Table 1 shows the data in one way of grouping, while Figure displays another. However, we prepared a new Figure.
Rev.: Table 2: This table shows the different test and cull strategies. None of the categories are based on removing all ELISA positives. Almost all require some other test to be done. This does not represent a uniform test and cull strategy that can be assessed.
AU: Yes, this is not a uniform strategy. The table shows the wide variance among the farms' own strategies when speaking about „test-and-cull” programs. Another question is whether it was really necessary to remove all ELISA-positive animals. According to the literature, it depends on the level of infection, the production, pregnancy status, etc, and farms can establish an order of decisions to wich animal should be culled first. All the strategies were established by the farm veterinarians based on their opportunity on their farms, including the budget they can spend on testing and culling animals. We restructured and simplified the Table.
Discussion
Rev.: Line 210-211: typo to be corrected.
AU: It is corrected.
Rev.: Line 212-215: Claims that the increase (in seroprevalence) found in the current study is likely to be lower than the actual Hungarian average as 27 of the farms had some sort of eradication programme for Ptb in place. This claim cannot be made as it is not stated how the study population differs from the national population in this respect and previously it claims that paratuberculosis is a greater problem in large herds and this study focused on large herds and accounted for “15% of the cattle population”
AU: This sentence was quite misguided, so we rephrased it.
Rev.: Lines 220-222: this statement cannot be supported by the findings as there was no evaluation of the adherence to the management aspect of the control programme and no common or rigorous approach to test and cull and therefore this study has not examined the effectiveness of either approach.
AU: We disagree with your statement. Although there were no uniform strategies for test-and-cull, we showed that farms that used both management and test-and-cull strategies achieved the best improvement in the apparent seroprevalence during the 4 years.
Rev: Lines 224:226: There is a claim that culling on production alone was the least effective strategy (and this is not a test and cull strategy), but there is no analysis to support this and there was only one of the study farms that took this approach.
AU: We reconsidered the classification of test-and-cull strategies, and the tables and text have been changed accordingly.
Rev: Lines 230-232: The approach of culling based on S/P value can be questioned and a more extensive evaluation of the recent literature on the subject is advised.
AU: I agree with you that culling on S/P can be questioned because the ELISA results can be influenced by the season, age, lactation period, prevalence of PTB in the herd etc., but if the farms can not afford to repeat the ELISA test or perform extra qPCR from faeces it could be a good choice to cull first the animals with high S/P value, because maybe they are the high shedders on the farm.
Rev: Lines 242-245: Here it states that “Compliance with management measures over the four-year study period has been unsuccessful in most cases.” This is problematic, because no information to support the degree of compliance has been reported in the study and secondly if compliance was generally poor (and it would appear to have been as judged by the variation in approach to test and cull) then it renders any comparison between the different categories meaningless.
AU: This text is deleted.
Rev.: Lines 249-250: “”makes immediate culling impossible”. This is not correct. Culling 5% of cows that are affected by a production limiting disease is unlikely to lead to financial ruin. It is certainly to be expected that herd managers might make such a claim to retain cows and in the short term that might result in higher milk sales. Such a claim cannot be made without some supporting evidence or argument.
AU: Culling of especially low-shedding animals would, however, imply premature culling, which can be very costly, especially in case of high milk yield of test-positive cows (Smith et al., 2009), not to mention that we do not cull pregnant animals (if possible). If the farms have other major disease problems in the herd, or reproduction-related problems leading to a lack of replacement animals, carrying out another control program eq. Staphylococcus aureus or BVDV eradication makes also it difficult to cull immediately even 5 % of the PTB-infected animals on the farm.
Rev.: Lines 252-257: Claims that using faecal PCR…. achieved a significant reduction of the Ptb prevalence within four years. No statistical analysis is presented to support this. There were seven farms that used PCR in various ways and in only one farm where PCR positives were culled after calving was the change in prevalence (-5%) different from other farms in the test and cull categories.
AU: True, but all the farms that used qPCR for the culling decision achieved greater improvement in the seroprevalence than the others that used only the ELISA method.
Conclusions
Rev.: Lines 272-276: These conclusions may be arrived by reviewing the literature, but the work done in this study does not allow them to be reached.
AU: Conclusions are rephrased.
Limitations of the study
Rev.: The limitations of the study are the lack of randomness in farm selection without any acknowledgement of the potential biases; and the failure to properly assess the degree of compliance to the chosen programmes at the farm level; or to assess the degree of success of the management programmes in achieving the objective of reducing the opportunity for the transmission of the disease.
AU: We supplemented the Limitations part.
Reviewer 5 Report
Comments and Suggestions for Authors
Brief summary:
The manuscript deals with the challenges of understanding and implementing the efficacy of PTB eradication programs in dairy herds.
The authors have carried out a study to investigate the prevalence of paratuberculosis in large-scale dairy herds in Hungary between 1 January 2018 and 31 December 2021, thus to compare the success of eradication/control programs on Hungarian farms, both with each other and with international literature. The overall aim of the study was to identify possible recommendations for farms in the establishment and implementation of PTB eradication programmes in dairy cattle.
The results of the study show that PTB eradication programmes must include compliance with biosecurity measures, particularly around calving and newborns. The 'test-and-cull' strategy alone is unlikely to lead to a reduction in PTB seroprevalence.
The study provides some elements to improve understanding of limitations of the current approach to the management and control of PTB.
Broad comments:
The reading of the manuscript is quite fluent and pleasant, I just have to point out some aspects that could be implemented:
- When the authors refer to 'hygiene management measures', they could directly introduce the concept of biosecurity, which has already been addressed in other studies similar to the present one.
Some of the following recent papers could be cited in this regard:
Imada, J. B., Roche, S. M., Thaivalappil, A., Bauman, C. A., & Kelton, D. F. (2023). Investigating Ontario dairy farmers motivations and barriers to the adoption of biosecurity and Johne's control practices. Journal of dairy science, 106(4), 2449–2460. https://doi.org/10.3168/jds.2022-22528
Scarpellini, R., Giacometti, F., Savini, F., Arrigoni, N., Garbarino, C. A., Carnevale, G., Mondo, E., & Piva, S. (2023). Bovine paratuberculosis: results of a control plan in 64 dairy farms in a 4-year period. Preventive veterinary medicine, 215, 105923. https://doi.org/10.1016/j.prevetmed.2023.105923
- When the authors describe the diagnostic methods available for PTB, they may refer to the WOAH Manual of Diagnostic Tests and Vaccines for Terrestrial Animals 2022;
WOAH. Chapter 3.1.16 Paratuberculosis (Johne’s Disease) (version adopted in May 2021). In Manual of Diagnostic Tests and Vaccines for Terrestrial Animals 2022; WOAH: Paris, France, 2022; Available online: https://www.woah.org/en/what-we-do/standards/codes-and-manuals/terrestrial-manual-online-access/
- In the Introduction paragraph, reference to at least European legislation and the PTB classification might be useful.
European Union Commission Implementing Regulation (EU) 2018/1882 of 3 December 2018 on the Application of Certain Disease Prevention and Control Rules to Categories of Listed Diseases and Establishing a List of Species and Groups of Species Posing a Considerable Risk for the Spread of Those Listed Diseases. Off. J. Eur. Union. 2018 L 308:21–29. Available online: http://data.europa.eu/eli/reg_impl/2018/1882/oj.
European Union Regulation (EU) 2016/429 of the European Parliament and of the Council of 9 March 2016 on Transmissible Animal Diseases and Amending and Repealing Certain Acts in the Area of Animal Health (‘Animal Health Law’) Off. J. Eur. Union. 2016 March 9;1–208.
- Furthermore, in the introduction paragraph, it might be useful to mention the gamma interferon test, a method that is currently used, only experimentally, for the early diagnosis of PTB, and that has proven to be useful in revealing infected individuals at an early stage of PTB.
The following articles could be referred to:
Corneli, S., Di Paolo, A., Vitale, N., Torricelli, M., Petrucci, L., Sebastiani, C., Ciullo, M., Curcio, L., Biagetti, M., Papa, P., Costarelli, S., Cagiola, M., Dondo, A., & Mazzone, P. (2021). Early Detection of Mycobacterium avium subsp. paratuberculosis Infected Cattle: Use of Experimental Johnins and Innovative Interferon-Gamma Test Interpretative Criteria. Frontiers in veterinary science, 8, 638890. https://doi.org/10.3389/fvets.2021.638890
de Silva, K., Begg, D. J., Plain, K. M., Purdie, A. C., Kawaji, S., Dhand, N. K., & Whittington, R. J. (2013). Can early host responses to mycobacterial infection predict eventual disease outcomes?. Preventive veterinary medicine, 112(3-4), 203–212. https://doi.org/10.1016/j.prevetmed.2013.08.006
Jungersen, G., Mikkelsen, H., & Grell, S. N. (2012). Use of the johnin PPD interferon-gamma assay in control of bovine paratuberculosis. Veterinary immunology and immunopathology, 148(1-2), 48–54. https://doi.org/10.1016/j.vetimm.2011.05.010
Specific comments:
Material and methods
Lines 119: Explain the meaning of Sample/Positive (S/P) values
Results
In Table 1, it would probably be better to insert the percentage “%” symbol next to the numbers.
In Table 2, it would be useful explain the meaning of Sample/Positive (S/P) values
Finally, it might be useful to represent the obtained data and results with graphical column plots.
Comments on the Quality of English Language.
Author Response
Reviewer 5
The manuscript deals with the challenges of understanding and implementing the efficacy of PTB eradication programs in dairy herds.
The authors have carried out a study to investigate the prevalence of paratuberculosis in large-scale dairy herds in Hungary between 1 January 2018 and 31 December 2021, thus to compare the success of eradication/control programs on Hungarian farms, both with each other and with international literature. The overall aim of the study was to identify possible recommendations for farms in the establishment and implementation of PTB eradication programmes in dairy cattle.
The results of the study show that PTB eradication programmes must include compliance with biosecurity measures, particularly around calving and newborns. The 'test-and-cull' strategy alone is unlikely to lead to a reduction in PTB seroprevalence.
The study provides some elements to improve understanding of limitations of the current approach to the management and control of PTB.
AU: Thank you for reading our manuscript, and for putting valuable comments and suggestions. Since the reviewers had many suggestions, we did not use the track changes function, instead, the changed parts are indicated in yellow.
Broad comments:
The reading of the manuscript is quite fluent and pleasant, I just have to point out some aspects that could be implemented:
Rev.: - When the authors refer to 'hygiene management measures', they could directly introduce the concept of biosecurity, which has already been addressed in other studies similar to the present one. Some of the following recent papers could be cited in this regard:
Imada, J. B., Roche, S. M., Thaivalappil, A., Bauman, C. A., & Kelton, D. F. (2023). Investigating Ontario dairy farmers motivations and barriers to the adoption of biosecurity and Johne's control practices. Journal of dairy science, 106(4), 2449–2460. https://doi.org/10.3168/jds.2022-22528
Scarpellini, R., Giacometti, F., Savini, F., Arrigoni, N., Garbarino, C. A., Carnevale, G., Mondo, E., & Piva, S. (2023). Bovine paratuberculosis: results of a control plan in 64 dairy farms in a 4-year period. Preventive veterinary medicine, 215, 105923. https://doi.org/10.1016/j.prevetmed.2023.105923
AU: Thank you. We cited the above papers.
- When the authors describe the diagnostic methods available for PTB, they may refer to the WOAH Manual of Diagnostic Tests and Vaccines for Terrestrial Animals 2022;
WOAH. Chapter 3.1.16 Paratuberculosis (Johne’s Disease) (version adopted in May 2021). In Manual of Diagnostic Tests and Vaccines for Terrestrial Animals 2022; WOAH: Paris, France, 2022; Available online: https://www.woah.org/en/what-we-do/standards/codes-and-manuals/terrestrial-manual-online-access/
AU: Thank you. We cited to the WOAH chapter.
Rev.: - In the Introduction paragraph, reference to at least European legislation and the PTB classification might be useful.
AU: Thank you. We inserted a reference to the Europeran legislation in the Introduction.
Rev.: - Furthermore, in the introduction paragraph, it might be useful to mention the gamma interferon test, a method that is currently used, only experimentally, for the early diagnosis of PTB, and that has proven to be useful in revealing infected individuals at an early stage of PTB.
AU: Thank you. We agree that the gamma interferon test has a potential in the diagnosis of PTB. However, with all due respect, we do not find it necessary to mention it in the manuscript since none of the examinations were gamma interferon tests, and our material is unrelated to this test.
Specific comments:
Material and methods
Rev.: Lines 119: Explain the meaning of Sample/Positive (S/P) values
AU: It is explained, and a new reference was inserted here.
Results
Rev.: In Table 1, it would probably be better to insert the percentage “%” symbol next to the numbers.
AU: Thank you for your suggestion. We would like to keep this structure. The table would overcrowded with the almost 30 „%” symbol. We used a new measurement unit, and this % is now pp (percentage point). Additionally, this table is now Table 2.
Rev.: In Table 2, it would be useful explain the meaning of Sample/Positive (S/P) values
AU: An explanation is inserted in the footnote of the table (which is now Table 3).
Rev.: Finally, it might be useful to represent the obtained data and results with graphical column plots.
AU: We inserted a new Figure with a strip chart.
Round 2
Reviewer 1 Report
Comments and Suggestions for Authors
Reviewer’s comments on R1 version.
The authors gave more specific detail in materials and methods as requested. As a whole, The changes in R1 version of the manuscript can be appreciated. The difficulties in more scientific analysis are understood.
In the “Conclusion” (L:369-370) the wording “This observational study……generated interesting hypotheses to be tested in future experiments” seems to be a very strong statement. This reviewer did not find such interesting hypotheses in the present manuscript. Please eliminate this part of the sentence or include and clarify the new (interesting) hypotheses appropriately.
Comments on the Quality of English LanguageMinor editing required
Author Response
Thank you for evaluating our manuscript.
We rephrased the conclusions and made a minor edit in English.
As the Editor requested, we also provided more information on the 'Conflicts of Interest' and 'Institutional Review Board Statement' parts.
Reviewer 4 Report
Comments and Suggestions for Authors
The authors have clarified important aspects of the methods used and the presentation of the results. As a consequence the conclusions drawn are now more nuanced.
Author Response
Thank you for evaluating our revision; we are glad you liked the fixes.